# Effects of Olive Oil on Blood Pressure: Epidemiological, Clinical, and Mechanistic Evidence

**DOI:** 10.3390/nu12061548

**Published:** 2020-05-26

**Authors:** Marika Massaro, Egeria Scoditti, Maria Annunziata Carluccio, Nadia Calabriso, Giuseppe Santarpino, Tiziano Verri, Raffaele De Caterina

**Affiliations:** 1National Research Council (CNR) Institute of Clinical Physiology, 73100 Lecce, Italy; Egeria.scoditti@ifc.cnr.it (E.S.); maria.carluccio@ifc.cnr.it (M.A.C.); nadia.calabriso@gmail.com (N.C.); 2Laboratory of Biochemistry and Molecular Biology, Department of Biological and Environmental Sciences and Technologies, University of Salento, 73100 Lecce, Italy; 3Cardiovascular Center, Paracelsus Medical University, 90471 Nuremberg, Germany; gsantarpino@gvmnet.it; 4GVM Care & Research, Città di Lecce Hospital, 73100 Lecce, Italy; 5Cardiac Surgery Unit, Department of Experimental and Clinical Medicine, University “Magna Graecia”, 88100 Catanzaro, Italy; 6Laboratory of Applied Physiology, Department of Biological and Environmental Sciences and Technologies, University of Salento, 73100 Lecce, Italy; tiziano.verri@unisalento.it; 7Institute of Cardiology, University of Pisa, 56126 Pisa, Italy

**Keywords:** hypertension, olive oil, monounsaturated fatty acids, polyphenols

## Abstract

The increasing access to antihypertensive medications has improved longevity and quality of life in hypertensive patients. Nevertheless, hypertension still remains a major risk factor for stroke and myocardial infarction, suggesting the need to implement management of pre- and hypertensive patients. In addition to antihypertensive medications, lifestyle changes, including healthier dietary patterns, such as the Dietary Approaches to Stop Hypertension (DASH) and the Mediterranean diet, have been shown to favorably affect blood pressure and are now recommended as integrative tools in hypertension management. An analysis of the effects of nutritional components of the Mediterranean diet(s) on blood pressure has therefore become mandatory. After a literature review of the impact of Mediterranean diet(s) on cardiovascular risk factors, we here analyze the effects of olive oil and its major components on blood pressure in healthy and cardiovascular disease individuals and examine underlying mechanisms of action. Both experimental and human studies agree in showing anti-hypertensive effects of olive oil. We conclude that due to its high oleic acid and antioxidant polyphenol content, the consumption of olive oil may be advised as the optimal fat choice in the management protocols for hypertension in both healthy and cardiovascular disease patients.

## 1. Introduction

Primary or essential hypertension can be currently defined, despite some variations across cardiovascular disease (CVD) societies, as the chronic elevation of blood pressure (BP) values over 130/80 mmHg, with no identifiable cause that, in the long-term, causes end-organ damage and results in increased morbidity and mortality [1]. In the last fifty years, the increasing access to safer and efficacious blood pressure (BP)-lowering drugs has increased longevity in primary hypertensive patients [2]. Nevertheless, hypertension remains the major promoter of the global incidence of CVD [3]. Since the World Health Organization now estimates that the number of hypertensive adults will increase from 1 billion to 1.5 billion worldwide by 2025 [4], a pressing need for a more efficient management of hypertensive patients is overwhelmingly arising in both developed and developing countries [4].

Beside genetic components [5], most of the pro-hypertensive environmental risk factors that contribute to disease progression match with wrong lifestyle habits [1]. They include an unbalanced dietary intake of sodium and potassium; excess of caloric intake, leading to overweight and obesity; and lack of physical activity. Together with conditions of persistent psychosocial stress and within the frame of predisposing genetic settings, these factors dynamically interact in determining a persistent elevation of BP [6]. Making proper dietary choices can be thus considered a strategic supportive tool in controlling BP [7].

## 2. Dietary Strategies in the Management of BP

Effective antihypertensive dietary strategies include reducing sodium intake, weight loss, limiting alcohol consumption, increasing potassium intake, and adopting an overall healthy eating pattern, such as the Dietary Approaches to Stop Hypertension (DASH) and the recently recognized potentially antihypertensive Mediterranean diet [8]. While DASH diet specifically encourages the consumption of fruits and vegetables; includes whole grains, poultry, fish, and nuts; and advises to reduce fats, red meat, sweets, sugar-containing beverages, and dairy products [9], “Mediterranean diet” (or, more properly, Mediterranean *diets*) is a more general descriptive term applied to dietary models traditionally adopted in several regions close to the Mediterranean Sea [10]. In general, Mediterranean diets are rich in plant foods including fruit; vegetables; bread; and other forms of cereal-derived foodstuffs, potatoes, beans, nuts, and seeds. Fruit represents the typical daily dessert, and processed sweets are only consumed weekly. Dairy products (mostly cheese and yogurt), fish, and poultry are consumed in low-to-moderate amounts; eggs are consumed weekly; red meat is consumed in low amounts, and wine in low-to-moderate amounts, usually with meals.

What substantially differentiates the Mediterranean diets from the DASH diet is its fat content: a low intake of saturated fats but a relatively high consumption of unsaturated fats, mostly deriving from olive oil (OO), which is used daily in stews, fried foods, salads and even desserts [11].

Already Ancel Keys, the father of Mediterranean diet, had identified OO as the key component of the healthful Mediterranean diet [12]. Promoting cardiovascular health, OO is recognized as a functional food [13]. However, the role of OO in the regulation of hypertension is still being studied. To preliminary screen the role of OO in BP management, we analyzed literature data on this topic enquiring PubMed with the following terms: “cardiovascular disease risk factors” and “Mediterranean diet” and “olive oil”. The resulted 239 papers were analyzed by the bibliometric mapping tool VOSviewer [14]. Bioinformatic analysis of medical subject headings (MeSH) associated to retrieved papers, returned a total number of 584 MeSH keywords, of which 68 met the threshold levels (minimum number of occurrence of a keyword = 10). In terms of “occurrence” the MeSH term “Mediterranean diet” resulted as the second cited key word, only preceded by the term “humans”. Next, we built up a term map based on the strength of the co-occurrence links with other keywords. The keywords with greatest total link strength were selected and highlighted as bubbles (Figure 1a). As shown in Figure 1b, among the retrieved MeSH keywords significantly connected to “Mediterranean diet” resulted: “hypertension”, “olive oil”, “unsaturated fatty acids” and “antioxidant”, which were therefore chosen as the search objects of the present review. The final aim was to evaluate whether the consumption of OO, in its major representative components, oleic acid and antioxidant polyphenols, may contribute to BP regulation in primary hypertensive patients and examine their potential mechanisms of action.

## 3. Olive Oil: A Succinct History, Composition, Production, and Consumption

OO has been long considered an extraordinary sacred and precious gift. The olive tree and OO are linked to the history and culture of some of the most ancient Mediterranean civilizations [15]. It is quoted in the Bible and in the Koran, mentioned by Homer in the Odyssey, and often present in Greek mythology [15]. The origin of the olive tree (*Olea europaea*) can be traced back to the Eastern and Middle East Mediterranean regions. From the Anatolian peninsula it spread out throughout the Mediterranean basin, initially by the Greeks (Figure 2a) and the Phoenicians, then by the Carthaginians, the Romans and the Arabs. In ancient Greece, athletes ritually rubbed it all over the body. OO was used to produce medicines and cosmetics (Figure 2b,c) [15]. Hippocrates called it “the great healer” and Homer “liquid gold”, while Galen praised it for its positive health effects [15].

At the time of the great transoceanic journeys, OO diffused to the Americas, and more recently, it has been introduced and cultivated in South Africa, China, Japan, and Australia [16]. Nowadays, OO is still mainly produced and consumed in the countries surrounding the Mediterranean Sea [17]. However, OO consumption in non-traditional markets has increased in the last decades, likely thanks to the growing appreciation of the Mediterranean diet outside the Mediterranean regions and to the increasing awareness of the role of OO and especially the extra-virgin olive oil (EVOO) in preserving the state of well-being and having a favorable effect against cardiovascular diseases [17]. EVOO is obtained from the fruit of the olive tree exclusively by mechanical or other physical means without any alteration or treatment other than washing, decantation or centrifugation, and filtration. These manufacturing techniques guarantee the transfer of minor bioactive components, including antioxidant polyphenols, from olives to OO [18].

The production of OO or EVOO rich in polyphenols requires considerable efforts compared to other types of oil production. The attainment of a good quality product, indeed, needs more complex agricultural and technological processes [15], which involve grater financial efforts. These factors probably curb the global production and consumption of EVOO, which represents only 2% of the total fat consumed worldwide, compared to 31% for soybean oil, 28% for palm oil, 13% for rapeseed oil and 9% for sunflower oil [19]. 

## 4. Olive Oil Composition

In bromatological terms, OO nutritional components are divided into two fractions: a major saponifiable and a minor non-saponifying (Figure 3). The most representative component is oleic acid, a monounsaturated fatty acid (MUFA), the proportions of which may vary from 55% to 83% of total fatty acids [20]. The minor components of EVOO represent only 2% of the total weight of OO and include various phytochemicals, such as triterpenes, sterols, tocopherols, carotenoids, and phenolic compounds [21] (Figure 3). These minor components represent, for the plant, a defense tool against the pro-oxidative environmental conditions typical of the Southern European latitudes [22,23]. The levels of these minor compounds are important parameters in characterizing quality, stability, and nutritional value of EVOO [21]. Recently, the OO phenolic components have attracted great scientific interest [24]. Most representative phenolic compounds include hydroxytyrosol (3,4-dihydroxyphenyletherol) and tyrosol (p-4-hydroxyphenyletherol) and their derivatives, such as the secoiridoids oleuropein (3,4-dihydroxyphenyletherol elenolic acid), oleocanthal, and oleacein [20]. The beneficial role of OO polyphenols has been attributed to their anti-atherosclerotic, anti-inflammatory and anti-oxidant properties [25]. Taking into account the positive results of a large number of clinical and population studies supporting the benefits of OO polyphenols, in November 2011, the European Food Safety Authority stated that “the polyphenols in OO contribute to the protection of blood lipids from oxidative stress”, suggesting a daily intake of 5 mg of OO polyphenols (20 g of EVOO) (http://www.efsa.europa.eu/) [26]

## 5. Olive Oil and Hypertension: Evidence from Epidemiological Studies

Historically, the first clear epidemiological evidence correlating fat consumption to BP regulation dates back to the end of the nineteen eighties, when Williams et al., in a small cross-sectional study conducted in 76 middle-aged US men, highlighted the existence of an inverse relationship between systolic and diastolic blood pressure (SBP, DBP) and oleic acid consumption as evaluated by 3-day food records [27]. These preliminary results, however, were not fully confirmed in subsequent larger prospective cohort studies.

In the Multiple Risk Factor Intervention Trial, for example, data from 24-h dietary recalls gathered from 11,342 middle-aged men were used to evaluate the relationships between the consumption of various dietary macronutrients and BP regulation. Data analysis showed a clear inverse association between DBP and the intake of polyunsaturated fatty acids (PUFA), as well as with the PUFA/saturated fatty acids (SFA) ratio; unexpectedly, however, the analysis failed to show a protective effect for MUFA [28]. In sharp contrast, in the Chicago Western Electric Company Study, led by J. Stamler, a cohort of 1714 middle-aged men was followed for 8 years to relate dietary habits to the annual change in BP. Among the several dietary variables included in the analysis, it turned out that all fats, including total, SFA, MUFA, PUFA and cholesterol, positively related to the average annual change in SBP but not DBP [29]. Apparently confirming Stamler’s results, among the 17,752 US participants in the Third National Health and Nutritional Examination Survey, those showing the highest values of SBP and DBP reported significant higher consumption of MUFA, PUFA, and cholesterol, thus confirming the apparent lack of any protective effects by these dietary fats [30].

The first cross-sectional study that investigated the association between specific fat consumption, including OO, and the incidence of hypertension in Southern Europe was conducted in Italy [31]. Briefly, the Italian Investigators from the Nine Communities Study evaluated the relationship between OO consumption, estimated by interviewer-administered dietary questionnaires and BP in 4900 middle-aged men and women with no previous diagnosis of hypertension. The results showed that OO consumption inversely correlated with SBP in both sexes and with DBP in men [31]. To this, data can be added from the largest population-based cohort study focused on the beneficial effects of the Mediterranean diet on cardiovascular health: this was the “Greek European Prospective Investigation into Cancer and Nutrition cohort (EPIC)” study [32]. Launched in the early twenties, it now encompasses about five hundred million people from ten European countries. Sub-studies in Greece, Italy, and Spain have concordantly found an inverse relationship between adherence to Mediterranean diet and BP. In the EPIC Greek cohort, the cross-sectional analysis of data gathered from 20,343 volunteers aged between 20 and 86 without previous diagnosis of hypertension showed that adherence to Mediterranean diet, and in particular the consumption of OO, besides vegetables and fruit consumption, was significantly and inversely associated with SBP and DBP in both sexes [33]. Of note, statistical adjustment between OO and vegetable intake, which are frequently consumed together, indicated that OO was the dietary component that exerted the dominant beneficial effect on BP [33]. From a practical point of view, after adjustments for sex, age, education, body mass index, waist-to-hip ratio, energy intake, physical activity, and consumption of vegetables, it turned out that for each 22 g increase in daily consumption of OO, SBP and DBP were lowered by 0.8 and 0.3 mmHg, respectively [33]. In the Italian cohort of EPIC, dietary and anthropometric data from 7601 women aged between 35 and 64 years with no previous diagnosis of hypertension were specifically analyzed with regard to SBP and DBP [34]. Again, the consumption of OO related negatively with DBP [34], while multivariate analyses, based on major dietary fatty acids subtypes (SFA, MUFA, and PUFA) suggested the existence of an inverse association between MUFA and DBP close to statistical significance [34]. The Seguimiento Universidad de Navarra-SUN-Study is another suggestive study performed in Southern Europe with the aim of investigating the relationship between OO and/or MUFA intake and the incidence of stroke, coronary events, hypertension, diabetes, and obesity in Spain [35]. In a preliminary evaluation of 4393 college students enrolled in the study, MUFA intake was associated with a lower prevalence of hypertension only in individuals with lower fruit and vegetable intakes, while the protective effect disappeared among individuals with higher fruit and vegetable consumption [36]. Furthermore, a prospective analysis of data from the SUN Study showed that OO consumption was inversely associated with the risk of developing hypertension only among men [37]. In this analysis, 5573 participants free of hypertension at baseline were followed for 28.5 months. Men in the highest quintile of OO consumption featured a 50% reduction in the risk of developing hypertension compared with those in the lowest quintile of consumption [37]. Here, the lack of association in women was interpreted as due to the low number of hypertensive cases in the female cohort [37].

These results agree with those that recently emerged from the International Study of Macro/Micronutrients and Blood Pressure (INTERMAP) trial, which evaluated the associations among SBP and DBP and an array of food and nutrient intakes in 4680 middle aged women and men enrolled in 17 geographically and culturally different cohorts in Japan, China, the United Kingdom, and the United States [38]. Here, data analysis demonstrated the existence of an inverse relationship between total MUFA and DBP in all participants. In particular, the estimated size effect was −0.8 mmHg and −1.7 mmHg for OO intake of 13 g/day, both in the entire cohort and in individuals not taking antihypertensive drugs [38]. Interestingly, for all populations investigated, the beneficial relationship was even stronger if accounting only oleic acid derived from vegetable sources [38].

Table 1 shows a summary of the information extracted from the included epidemiological studies.

## 6. Olive Oil and Hypertension: Evidence from Clinical Trial. Search Strategy

Although from observational studies many positive evidences have emerged linking OO to lower BP, the modest validity of nutritional findings derived from research methods adopted [39] have pushed nutritional investigators to improve confidence in olive oil-related nutritional outcomes, undertaking evidence-based approaches relying on randomized clinical trials (RCTs). RCTs, indeed, are the gold standard for determining a causal relationship between an exposure and an outcome, and therefore, they are reviewed as benchmark for medical guidelines productions and health-care policy [40]. Minimizing or eliminating confounding bias, which are the major challenges in observational studies, RCTs are increasingly used to examine the effect of a nutrient or food on cardiovascular events, in a fashion identical to that required for approving new drugs and medical devices [40]. For this reason, we conducted a comprehensive search in PubMed, Scopus, and Web of Science to identify the RCTs, investigating the management of BP by OO and its major constituents (oleic acid and antioxidants polyphenols).

To identify and retrieve all potentially relevant articles on this topic, we performed the search utilizing the following expressions: “olive oil” AND “arterial pressure” OR “hypertension” OR “blood pressure” OR “diastolic pressure” OR “systolic pressure” AND “randomized controlled trial” AND “humans”. We searched for articles published in each database from its inception until September 2019. We also performed additional manual search by analyzing the reference list of original publications, reviews and meta-analyses on the same topic. In the final selection, we included and discussed 19 papers specifically addressing the role of OO in healthy subjects and/or in patients at cardiovascular risk, including hypertensive and diabetic subjects. Table 2 shows a summary of the information extracted from the included studies. Study country origin were: Spain [41,42,43,44,45,46,47,48], Denmark [49,50], Italy [51], the UK [52], the USA [53,54], Brazil [55], and Germany [56], while two studies were multinational and included Finland, Denmark, Germany, Italy, and Spain [57], as well as Finland, Denmark, Italy, Australia, and Sweden [58]. The 19 studies overall evaluated 8247 adult participants without cardiovascular events. Many participants were hypertensive [46,47,48,51,53,56] or had hypertension in association with other cardiovascular risk factors [42,43,45]. Two studies were performed in overweight or obese patients [52,54], while 4 studies enrolled subjects with diabetes [41,49,50] or with metabolic syndrome [55]. Finally, in 3 studies, healthy subjects were enrolled [44,58,59]. OO was mostly administered in liquid form as EVOO, from 10 to 60 mL daily [41,42,43,44,50,51,54,55] with different dietary background, including both unhealthful eating habits (including the classical Western diet) [54] or the supposedly healthful Mediterranean and DASH diets [41,42,43,44,46,53]. In some studies, OO was also provided in capsules, providing 1–6 g of OO daily [52,56] or as oleic acid-enriched margarine [58]. Since repeated measurements of BP more accurately reflect BP levels than single measurements, in both clinical practice and hypertension research, the 24-hour ambulatory blood pressure monitoring (ABPM) is considered the gold standard for examining the influence of pharmacological and nutritional interventions on BP [60]. However, among the 19 studies selected, only 3 evaluated BP by ABPM [43,49,50]. Finally, apart from some mild gastrointestinal adverse reactions, no important adverse events were in general associated with OO consumption. 

## 7. Olive Oil and Hypertension: Clinical Trial Results

From a chronological perspective, one of the earliest randomized clinical studies assessing the antihypertensive effect of OO dates back to the late 1980s [59]. Here, the antihypertensive effect of a high-fat diet enriched in OO was compared to that of a reference diet, low in fat and rich in carbohydrate in a sample of 47 healthy subjects. After 36 days, SBP and DBP were significantly reduced in both experimental arms, with no significant difference between tested diets, thus suggesting, for the first time, a chance of regulating BP by manipulating the amount of dietary OO [59]. Some years later, these results were confirmed [50] in a group of 15 type-2 diabetic subjects, in which even more powerful antihypertensive effects by a diet enriched in OO as compared with a high-carbohydrate diet were observed [50]. The same authors, some years later, compared the effects of 3-weeks administration of an OO-enriched diet to an isocaloric diet enriched with grapeseed oil (a source of the PUFA linoleic acid) [49]. Although the effects in terms of modulation of blood glucose and lipid levels were similar for the two groups, the OO-enriched diet exhibited grater antihypertensive effects than grapeseed oil-enriched diet [49]. In the wake of these encouraging results, some years later, the use of OO as a supportive nutraceutical tool in the drug-management of hypertension was specifically evaluated [51]. In this provocative randomized cross-over trial, 23 hypertensive patients were assigned to a diet rich in EVOO or a diet rich in sunflower oil (SO). After 6 months, only the OO diet significantly lowered SBP and DBP and, of note, the daily dosage of antihypertensive drugs needed to curb BP was significantly reduced only upon OO-enriched diet [51], thus proposing OO as a potential nutraceutical in the adjunctive treatment of hypertension [51]. 

From this study forward [51], the number of clinical reports addressing the role of OO in the management of hypertension has risen steadily. By a cross-over randomized approach, between the 2003 and 2005, the OmniHeart collaborative research group compared the effect of three different healthful diets on BP and serum lipids in a group of 164 pre-hypertensive adults not taking antihypertensive medications. Tested diets were a diet rich in MUFA as derived from olive, canola, and safflower oils; a carbohydrate-rich diet (like the DASH diet); and a protein-rich diet. After 6 weeks, as compared with the carbohydrate-rich diet, both MUFA- and protein-rich diets reduced significantly BP and positively regulated the lipid profile [53]. These results were confirmed also in healthy subjects [58]. In this trial, 162 subjects were randomly assigned to one of the following diets: a diet enriched in MUFA and a diet enriched in saturated fats. The MUFA diet included butter and margarine containing high amounts of oleic acid derived from OO and SO, while the SFA diet included butter and margarine containing high proportion of SFAs from an unspecified food source. Each group was further randomly divided to receive fish oil or placebo. After 3 months, both SBP and DBP decreased with MUFA but did not change with SFA. Interestingly, the addition of fish oils did not improve the effect of the MUFA diet any further [58]. The potential of EVOO as antihypertensive tool has been also demonstrated in a sample of hypertensive elderly people [48] exposed daily to 60 g EVOO or SO. After 1 month, SBP values were normalized only in the EVOO group [48]. Interestingly, similar positive findings were also observed in a sample of 20 elderly overweight and obese subjects, randomized to include, within a Western-diet regimen, EVOO, or corn oil and soybean oil, for 3 months [54]. The study, however, that definitively confirmed the usefulness of OO in the management of BP is the “Prevención con Dieta Mediterránea (PREDIMED)” trial [61], in the sub-analyses performed by Toledo [42] and Domenech [43]. The PREDIMED study has been carried out in two Spanish centers and involved more than 7000 subjects at high cardiovascular risk and with hypertension. Here subjects were randomized to one of the following three diets: a Mediterranean diet supplemented with EVOO, a Mediterranean diet supplemented with nuts, or a control diet in which subjects were only advised to reduce dietary fat [61]. The main outcome was an aggregate of non-fatal myocardial infarction, non-fatal stroke, or cardiovascular death [61]. After a follow-up period of 4.8 years, DBP decreased by 1.5 and 0.65 mmHg in the EVOO group and in the nuts group, respectively [42], while no significant effects on SBP were highlighted. However, in the sub-study of Domenech [43], in which BP was evaluated by the more accurate ABPM method, in addition to DBP, a significant lowering effect on SBP was evident in both the EVOO- and nut groups. 

In recent years, among the minor constituents of OO potentially responsible for the antihypertensive effect, the role of the antioxidant polyphenols has been largely assessed. This was first assessed in a sample of 40 coronary Spanish male patients [45]. Here, participants (half of whom with hypertension) were cross-randomized to receive polyphenol rich EVOO or refined OO (containing very low amount of antioxidant polyphenols) for 3 weeks. The authors observed that the intake of both oils reduced SBP in hypertensive patients. However, the effect was stronger with a polyphenol-rich EVOO, suggesting a specific role for OO antioxidants in the mechanisms of BP regulation [45]. These protective effects were confirmed some years later also in a mixed group of healthy male and female subjects [44]. More recently, 24 normotensive and hypertensive women (not formerly taking antihypertensive or lipid-lowering drugs) were randomized to receive a Mediterranean diet supplemented with a polyphenol-rich OO or a Mediterranean diet supplemented with polyphenol-free OO [46]. After 2 months, a significant decrease in SBP and DBP was evident only with polyphenol-rich OO. Interestingly, the extent of BP reduction observed in this study was similar to that obtainable upon commonly prescribed first-line antihypertensive drugs [46]. We acknowledge, anyway, that not all studies have shown beneficial effects of MUFA or OO on BP, especially when OO was administered as encapsulated extracts [52,56,62].

## 8. Pathogenesis of Hypertension

Being BP the product of cardiac output and systemic vascular resistance, hypertension may follow from an increase in cardiac output, an increase in systemic vascular resistance, or both [1]. Under normal physiological conditions, BP homeostasis requires the coordinated interplay of several neuro-humoral and cellular players, that include the renin-angiotensin-aldosterone system (RAAS), the sympathetic nervous system, natriuretic peptides, the endothelium, and smooth muscle cells [1]. Under condition of low renal blood flow, juxtaglomerular cells in the kidneys are primed to convert plasma pro-renin into renin. Renin mediates the conversion of angiotensinogen, largely released by liver, to angiotensin (Ang) I. Ang I is subsequently converted to Ang II or to a series of other vasoactive angiotensin-related peptides by angiotensin-converting enzymes (ACEs), which are expressed by the endothelial cells of all blood vessels, in large amounts within the lungs, and by epithelial cells within the kidneys [64]. Ang II is regarded as the key player in BP system regulation [65]. It exerts potent vasoconstrictive effects, causing blood vessels narrowing and stimulating the secretion of aldosterone from the adrenal cortex. This latter acts on the tubules in the kidneys, increasing the reabsorption of sodium and water and the excretion of potassium, thus determining the BP increase [66], but also inducing, together with Ang II, endothelial dysfunction [67]. Ang II also causes the release of the anti-diuretic hormone vasopressin [64]. Produced by the hypothalamus and released from the posterior pituitary gland, vasopressin exerts, as the name indicates, vasoconstrictive effects, stimulating the reabsorption of water in the kidneys, acting on the central nervous system, increasing the appetite for salt and stimulating the sensation of thirst [66] (Figure 4). Over time, derangements of any of these regulatory systems, if leading to increased BP for a sufficient long time, may increase vascular stiffness and promote atherosclerosis, target-organ damage, and acute cardiovascular events [1]. Persistent conditions of psychosocial stress, excessive dietary sodium intake, sedentary lifestyles, overweight and obesity are recognized among the most common environmental risk factors able to interact within a predisposing genetic setting, in determining the elevation of BP [1]. In particular, the hypertrophic adipose tissue produces and releases several pro-hypertensive molecules, such as free fatty acids, leptin, angiotensinogen, pro-inflammatory cytokines, and reactive oxygen species (ROS) [6]. These molecules increase BP affecting the vessel wall, the brain, and the kidneys, leading to combinations of augmented vasoconstriction, reduced vasodilation, fluid retention, and/or increased vascular stiffness [68]. Furthermore, since the prevalence of high BP increases with age, hypertension is also considered a typical condition of aging [69] (Figure 4).

Pathogenically, functional and structural changes eventually lead to increased vascular stiffness and atherosclerosis in the hypertensive arterial wall, including endothelial dysfunction, abnormal vascular smooth muscle cell (VSMC) growth and migration, inflammation, fibrosis, altered contractility, and vascular hypertrophic remodeling (Figure 4). The underlying humoral changes encompass a reduced production of vasodilators such as nitric oxide (NO) and prostacyclin, increased expression of pro-fibrotic and pro-inflammatory molecules, and increased expression and activity of matrix metalloproteinases (MMPs) that prime the shift of VSMCs and endothelial cells towards proliferative, hypertrophic, and pro-inflammatory phenotypes [70] (Figure 4).

Interestingly, many pro-hypertensive stimuli, including Ang II, aldosterone oscillatory shear stress, as well as pro-inflammatory cytokines and growth factors, exert their pro-hypertensive vascular effects through nicotinamide adenine dinucleotide phosphate (NADPH) oxidase-derived ROS [71]. At the cellular level, ROS can neutralize the vasodilator NO [72], inactivate protein tyrosine phosphatases, increase intracellular free calcium concentrations, and act as second messengers within redox-dependent signaling pathways, including p38 MAPK, ERK1/2, ERK5 and RhoA/ROCK, to induce cell migration and proliferation, as well as the activation of redox-sensitive pro-inflammatory and pro-angiogenic transcription factors, including nuclear factor(NF)-κB, activator protein(AP)-1, retinoic acid receptor, and hypoxia-inducible factor-1 [73].

## 9. Olive Oil Anti-Hypertensive Effects: Mechanistic Evidence from Animal and Human Studies

In spontaneously hypertensive rats (SHR), both BP and the vascular expression of the redox-sensitive transcription factors NF-κB and AP-1 were shown to be downregulated by a plethora of synthetic antioxidants [74], suggesting the potential for EVOO to affect hypertension through anti-oxidative mechanisms. In line with this hypothesis, in the same SHR model, the administration of EVOO [75], pomace oil [76], or related antioxidant polyphenolic components (oleuropein, hydroxytyrosol) was shown to lower BP significantly. Investigated mechanisms range across anti-oxidative activities [75,77], changes of aminopeptidase activities favoring the production of Ang 2-10 (in this way compensating the activity of Ang II) and the clearance of Ang III and Ang IV [75], increase in the endothelial nitric oxide synthase (eNOS) expression [76,78], and reduction of plasma Ang II [75] (Figure 4). Correspondingly, our previous research has shown that OO polyphenols and related human serum metabolites, as found in human serum upon EVOO ingestion, can decrease the endothelial activity and expression of NADPH oxidase, as shown by the reduced levels of intracellular ROS and decreased expression and activity of MMP-2 and MMP-9 in endothelial cells [79,80] and adipocytes [81,82].

Although all the aforementioned antihypertensive mechanisms have been attributed to the potential interference of EVOO antioxidant components within the redox-signaling pathways leading to hypertension [74], interesting antihypertensive mechanisms have also been proposed for the fat component oleic acid [83]. In SHR aorta, it was shown that the increase in oleic acid within cellular phospholipids downregulates the expression of a series of G protein-coupled receptors, including the adrenoceptor alpha 2A, the G protein subunit alpha i2, the G protein subunit alpha i3, the G protein subunit alpha 11, and phospholipase C beta, which inhibit adenylyl cyclase activity [83,84]. In the same experimental setting, analogs of oleic acid, elaidic and stearic acids, did not exhibit hypotensive activity, indicating that the molecular mechanisms linking oleic acid to BP regulation are connected to membrane lipid structure in a highly specific fashion [83].

Accordingly, in a subsample from the EUROLIVE study [85], the administration of a polyphenol-rich EVOO to healthy individuals was shown to reduce the expression of genes encoding for ACE and adrenoceptor beta 2 [63], while in a subsample from the PREDIMED study, 1-year EVOO supplementation increased serum levels of NO and the peripheral cell expression of eNOS [86]. The molecular mechanisms by which OO components influence gene expression are not fully understood. Both oleic acid [87] and antioxidant phenolic compounds oleuropein and hydroxytyrosol [88,89] interact with cellular signaling cascades regulating the activity of NF-κB and AP-1 and, consequently, affect the expression of the respective targeted genes [90]. Moreover, interaction of phenolic compounds with miRNAs should be taken into account as another potential molecular mechanism for OO activities [91]. Recent data have indeed suggested interesting roles for microRNAs in the regulation of RAAS, either as mediators or for being targets of RAAS pharmacological inhibitors [92]. Interestingly, newly appreciated olive phenolic compounds, including the secoiridoids oleocanthal and oleacein, besides being able to stabilize atherosclerotic plaques in hypertensive patients by modulating the expression MMPs [93], have shown the ability to modulate the expression of several miRNAs connected to the NF-κB pathway [94] (Figure 4). 

## 10. Discussion

Since the first use of antihypertensive agents in the Veterans Administration Cooperative Study in the late 1960s [95], the increasing access to BP lowering medications has improved longevity and quality of life in hypertensive patients [2]. Nevertheless, hypertension still remains a major risk factor for stroke and myocardial infarction, suggesting the need to implement BP screening programs to ensure early diagnosis finalized to a proper management of hypertensive patients.

As a consequence of recent studies showing benefits associated with more pronounced reduction of BP in the hypertensive population, the American College of Cardiology and American Heart Association Task Force have updated the guidelines for the management of BP in adults [96]. Besides setting the BP cut-off values to the current 130/80 mmHg levels, they expressly recommended the adoption of proper lifestyle measures in all hypertensive patients, including those who require drug treatment, since the non-adoption of lifestyle changes can blunt the therapeutic effect of antihypertensive medications and prevent the achievement of BP target values [96]. The systematic analysis of the role of nutrients in BP management has therefore become mandatory in the era of “evidence-based nutrition” [97,98].

A great deal of population-based nutritional studies have investigated and recognized several dietary components as promoting or protecting from hypertension [99]. Among protective factors, a significant role has been postulated for a typical component of the cardiovascular-protective diet par excellence, the Mediterranean diet [100]. In this review, we have systematically analyzed the effects of OO consumption and its major components, oleic acid and antioxidant polyphenols, on BP management in healthy and at-risk subjects, and examined the underlying mechanisms of action.

To this aim, we first reviewed all epidemiological evidence investigating the association between MUFA and/or OO and the incidence of hypertension. In agreement with Alonso and coworkers [101], we observed that results were strictly dependent on the country where studies were performed. In particular, we observed that epidemiological surveys performed in the USA and, presumably, in all countries adopting Western-like dietary patterns failed to show clear protective effect for MUFA or OO assumption (Table 1), while a protective effect by OO and MUFA was evident in studies performed in Mediterranean countries (Table 1). Although all these studies adjusted their analyses for confounding factors, the emerging lack of a clear protective effect by MUFA in non-Mediterranean countries should not be surprising: such studies were conducted in countries where the overall MUFA consumption is moderate and where there is a high consumption of meat, itself associated with a high intake of SFA [102]. Conversely, epidemiological studies enrolling dwellers in Southern Europe yielded very different results, proposing a clear protective role for MUFA and OO in the regulation of BP.

In agreement with Nissensohn [30], more concordant results were obtained under the controlled conditions ensured by RCTs. By examining the selected 19 studies specifically addressing the role of OO in healthy and vascular disease conditions, including hypertensive and diabetic subjects, we observed that sub-chronic and chronic intake of OO, especially if ranging from 10 to 60 mL of daily intake of EVOO, consistently reduced SBP and, to a lesser extent, DBP (Table 2). The extent of BP reduction was, however, rather variable, ranging between −10 and −1 mmHg for both SBP and DBP (Table 2). It has been estimated that a 3 mmHg average reduction in SBP could lead to an 8% reduction in stroke mortality and a 5% reduction in mortality from ischemic heart disease [103]. Thus, the apparently modest small BP reduction induced by regular intake of EVOO may translate in enormous beneficial impacts.

The heterogeneity in BP changes found across studies may depend on various factors, including intervention or exposure duration, OO amount and quality, and dietary and genetic background. For example, the CLOCK single nucleotide polymorphisms (SNPs) rs4580704 C > G has been associated with obesity, hyperglycemia, and hypertension [104]. Interestingly, carriers of the minor allele (G), besides having a lower risk of hypertension as well as lower body weight, fasting-insulin, and hyperglycemia [104], appear to be more protected if following a Mediterranean diet [105] and/or when their MUFA intake was >13.2% of total energy [104].

Therefore, to decrease the within-group heterogeneity in terms of outcome, some important issues such as genetic stratification of hypertensive patients [106] as well as timing and doses of OO intake remain to be established and should represent the focus of future research. At this time, on a preliminary base and taking into account the literature here revised, a good choice to combat hypertension should include the regular use of an EVOO containing high amount of antioxidant polyphenols, in the order of at least 5 mg of bioactive phenols in 20 g of OO daily (2 heaped spoons) [26].

Advances in understanding of the complex cellular and molecular pathophysiology of hypertension [1] have been of help to set-up a rational use of dietary strategies in the management of hypertension [1]. In particular, cell culture experiments, animal model of hypertension, and human metabolic studies have provided consistent mechanistic data demonstrating a role for altered redox-signaling mechanisms in the pathogenesis of cardiovascular disease and hypertension [71]. EVOO is a natural fruit product that contains a unique composition of oleic acid and minor constituents, most of which feature marked antioxidant properties [107]. Data from both human and animal studies concordantly show BP lowering effects of oleic acid and minor constituents from EVOO through different mechanisms (Figure 4). By regulating phospholipid architecture, the OO oleic acid appears to improve membrane functions and cell physiology [83]. On the other side, both OO and antioxidant polyphenols regulate the release of vasoconstrictors, vasodilators, and pro- and anti-inflammatory molecules through genomic mechanisms [70].

## 11. Conclusions

Both experimental and human studies are overall robust in showing anti-hypertensive effects by OO, due to its high oleic acid and antioxidant polyphenols content. On the basis of reviewed results, it is difficult to establish which is the component that most efficaciously contributes to BP regulation. Rather, the whole range of EVOO components appears to express their maximal efficaciousness within the complex of their original matrix. This suggests that formulation of possible functional products should approximate as much as possible the natural environment in which active molecules are found. EVOO thus appears to be the optimal fat choice in management protocols for hypertension in both otherwise healthy and cardiovascular patients.

## Figures and Tables

**Figure 1 nutrients-12-01548-f001:**
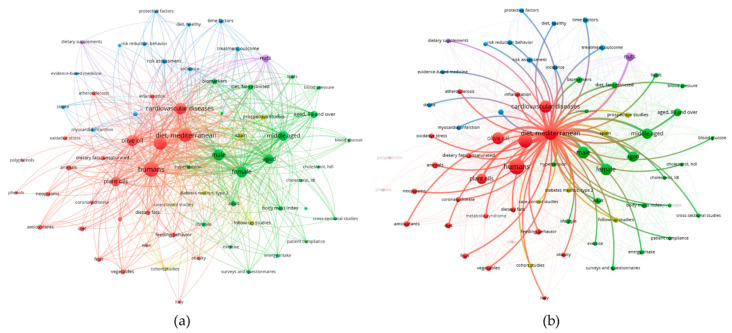
(**a**) The bubble map visualizes the medical subject headings (MeSH) keywords selected from papers published and retrieved in PubMed under the search terms term “cardiovascular disease risk factors” and “Mediterranean diet” and “olive oil”. The bubble size indicates the frequency of occurrence of the words, while the bubble color represents the cluster of belonging. Two bubbles are in closer proximity if the two words had more frequent co-occurrence. (**b**) The map highlights MeSH terms directly connected to Mediterranean diet. Analysis was performed by the bibliometric mapping tool VOSviewer.

**Figure 2 nutrients-12-01548-f002:**
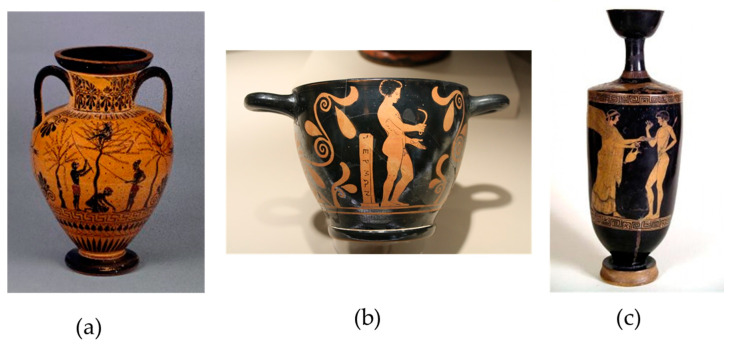
(**a**) Black-figured Greek amphora showing a scene of olive-gathering. A naked youth seated in a tree shakes down olives with sticks. Two bearded figures beat the trees with sticks, and a naked youth collects the fallen olives in a basket. The amphora dates back to 520 BC. British Museum, London. (**b**) Greeks and Romans used OO to clean their bodies after exercise. They smeared OO on the body so that it might collect dirt and sweat and then scraped it off using a metal instrument called a strigil. The red figure on the cup (skyphos) depicts a nude athlete holding a strigil. The skyphos dates back to 410 BC, Archaeological Museum, Milan. (**c**) Lekythos, vase used as a container of olive oil for body care of athletes. This lekythos dates back to 500 BC. Archaeological Museum, Bologna.

**Figure 3 nutrients-12-01548-f003:**
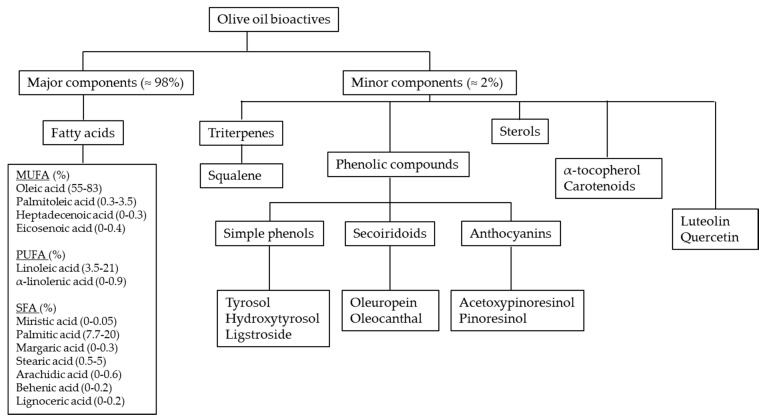
Main bioactive compounds in extra-virgin olive oil (EVOO).

**Figure 4 nutrients-12-01548-f004:**
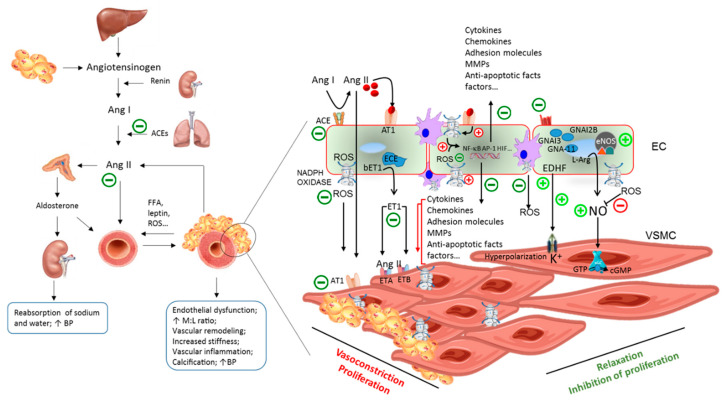
A model describing interactions of OO components—oleic acid and antioxidant polyphenols—within the pathogenic process leading to hypertension. Ang, angiotensin; FFA, free fatty acids; ROS, reactive oxygen species; BP, blood pressure; M:L ratio, media to lumen ratio; ACEs, angiotensin converting enzymes; ECE, endothelin converting enzyme; bET1, big endothelin-1; AT1; Angiotensin II receptor type 1; ETA, Endothelin receptor type A; ETB, Endothelin receptor type B; MMPs, metalloproteinases; NO, nitric oxide; eNOS, endothelial nitric oxide synthase; EC, endothelial cell; VSMC, vascular smooth muscle cell; GTP, guanosine-5′-triphosphate; cGMP, cyclic guanosine monophosphate.

**Table 1 nutrients-12-01548-t001:** Epidemiological studies evaluating the relationship MUFA intake, OO consumption and hypertension.

First Author, Year [Ref]	Country	Design	Participants	Sex	Main Results
Williams, 1987, [1]	USA	Cross-sectional	76	male	MUFA ↓ SBP and DBP
Stamler, 1996 [2]	USA	Cohort	11,342	male	MUFA ↔ SBP and DBP
Stamler, 2002 [3]	USA	Cohort	1714	male	MUFA ↑ SBP
Hajjar, 2004, [4]	USA	Cross-sectional	17,752	male and female	MUFA ↑ SBP and DBP
Trevisan, 1990 [5]	Italy	Cross-sectional	4903	male and female	OO ↓ SBP
Alonso, 2004 [6]	Spain	Cohort	6863	male and female	OO ↓ hypertension risk in men
Psaltopoulou, 2004 [7]	Greece	Cross-sectional	20,343	male and female	OO ↓ SBP and DBP
Masala, 2008 [8]	Italy	Cross-sectional	7601	female	OO ↓ DBP
Miura, 2013 [9]	China, Japan, UK, USA	Cross-sectional	4680	male and female	MUFA ↓ DBP

Abbreviations: OO, olive oil; MUFA, monounsaturated fatty acids; BP, blood pressure; SBP, Systolic blood pressure; DBP, diastolic blood pressure. ↓ = downregulation; ↑ = upregulation; ↔ = no effects.

**Table 2 nutrients-12-01548-t002:** RCT assessing the effect of olive oil on blood pressure.

First Author, Year [ref]	N, Sex, Age (yr), Weight (kg) (or BMI)	Health Status	Study Design and Country	Intervention	Measure	Administration	Δ SBP (mmHg)	Δ DBP (mmHg)	Main Results
Mensink, 1988 [59]	47, male and female, 27 ± NR; 71 ± NR	healthy	RCT, parallel, Netherlands	8-week OO-enriched diet vs. CHO diet	Office	liquid	−2.3 in CHO-group; −2.7 in OO-group	−4.7 in CHO group; −4.4 in OO-group	Both interventions decreased SBP and DBP significantly, but there were no differences between groups
Rasmussen, 1993 [50]	15, male and female, 57 ± 2; 80.5 ± 3.8	diabetics	RCT, cross-over, Denmark	3-week intervention with 3-week wash-out period, EVOO-enriched diet vs. CHO diet	24-AMBP	liquid, cold pressed olive oil	−4.0 in EVOO diet vs. CHO diet	−3.0 in EVOO diet vs. CHO diet	SBP and DBP significantly lower after MUFA diet than CHO diet
Rasmussen, 2006 [58]	162, male and female, 48.5 ± 8.0, 26.5 ± 3.8 kg/m^2^ (BMI)	healthy	RCT, parallel, multicenter	12-week intervention with MUFA diet or SFA diet. Each group was further randomly assigned to receive supplementation with fish oil or placebo	Office	margarine with a high proportion of oleic acid, derived from high-oleic acid sunflower oil	−2.2%	−3.8%	SBP and DBP decreased with the MUFA diet but did not change with the SFA diet
Thomsen, 1995 [49]	16, male and female, 59 ± 7, 81.6 ± 15.1	diabetics	RCT, cross-over, Denmark	3-week intervention with 3-week wash-out period, MUFA diet (EVOO) vs. PUFA diet	24-AMBP	liquid, cold pressed olive oil	−5.1 in EVOO diet vs. PUFA diet	−3.8 in EVOO diet vs. CHO diet	SBP and DBP were lower after EVOO diet than after PUFA diet
Ruiz-Gutierrez, 1996 [47]	16, female, 56.2 ± 5.4, NR	NC and HC hypertensive women	RCT, parallel, Spain	4-week intervention with 4-week wash-out period, EVOO vs. HOSO	Office	liquid, cold pressed OO or SO	−10 in NC women on EVOO; −7 in HC women on EVOO; −6 in NC women on HOSO; −2,26 in HC women on HOSO	−10 in NC women on EVOO; −6 in HC women on EVOO; −4 in NC women on HOSO; −5.92 in HC women on HOSO	Significant decrease in SBP and DBP after EVOO but not after HOSO
Ferrara, 2000 [51]	23, male and female, age range 25–70, 70 ± 9	hypertensive	RCT, cross-over, Italy	24-week intervention, EVOO diet vs. SO diet	Office	liquid, cold pressed OO or SO	−7 in EVOO diet; +1 in SO diet	−6 in EVOO diet; no change in SO diet	SBP and DBP decresed after EVOO but not after SO. Reduced need for antihypertensive drugs after EVOO
Appel, 2005 [53]	164, male and female, 53.6 ± 10.9; 87.3 ± 18.7	pre-hypertensives and hypertensives	RCT, cross-over, USA	CHO-rich diet (similar to the DASH trial) vs. protein-rich diet vs. MUFA-rich diet-2-4-week wash-out period between each feeding period.	Office	liquid, olive, canola, and safflower oils beside to nuts and seeds	−9.3 in MUFA rich diet; −8.2 in CHO rich diet	−4.8 in MUFA rich diet; −4.1 in CHO rich diet	SBP and DBP were lower after MUFA-rich diet compared with CHO diet; no significant difference there were between protein and MUFA diets
Perona, 2004 [48]	62, male and female, 84.0 ± 7.4; 28.8 ± 5.2 kg/m^2^ (BMI)	hypertensive and normotensive	RCT, cross-over, Spain	4-week intervention with 4-week wash-out period, EVOO diet vs. SO diet	Office	liquid, cold pressed OO or SO	−12 in EVOO diet vs. SO diet in hypertensive	no difference between EVOO and SO	Normalization of SBP after EVOO in hypertensive individuals. No effect on DBP
Taylor, 2006 [52]	40, men, 47 ± 8; 97 ± 13	overweight	RCT, parallel, UK	OO, 6 g/day capsules; CLA, 4.5 g/day capsules	Office	capsules	+0.2 in OO group; −0.4 in CLA group	−0.8 in OO group; +0.1 in CLA group	With OO only DBP decrease (by trend); with conjugated linoleic acid only SBP decrease by trend
Konstantinidou, 2010 [44]	90, male and female, 45 ± 10; 68 ± 15	healthy	RCT, parallel, Spain	12-week intervention with MD + OO (low polyphenol content) or MD +EVOO (high polyphenol content); habitual diet	Office	liquid	−1.63 in low polyphenol OO; −0.4 in EVOO; +1.4 in control diet	−0.8 in low polyphenol OO; +1.12 in EVOO +1.67 in control diet	With low polyphenols OO SBP and DBP decrease (by trend); with EVO only SBP decrease (by trend)
Fitò, 2005 [45]	40, male, 68 ± 8; 27.5 ± 3 kg/m^2^ (BMI)	at CHD risk and hypertensive	RCT, cross-over, Spain	3-week intervention with 2-week wash-out period, EVOO vs. ROO	Office	liquid	−2.53 in EVOO group vs. ROO group	+1.16 in EVOO group vs. ROO group	With EVOO SBP decreased in hypertensive patients. No changes were observed in DBP
Moreno Luna, 2012 [46]	24, women, age range: 24–27 years, BMI range: 23.5–27.1 kg/m^2^	hypertensive and normotensive women	RCT, cross-over, Spain	8-week intervention with MD + OO (polyphenol free) or MD + EVOO (rich in polyphenols); 4-week wash-out period;	Office	liquid	−7.91 in EVOO group vs. baseline; −1.65 in OO polyphenol free group vs. baseline	−6.65 in EVOO group vs. baseline; −2.17 in OO polyphenol free group vs. baseline	Only polyphenol-rich OO decrease SBP and DBP
Bondia-Pons, 2008 [57]	160, male, 33.3 ± 11.1; 75.8 ± 9.7 kg	healthy	RCT, cross-over multicenter	3-week intervention with 2-week wash-out periods. OO with different polyphenol content (low, medium, high)	Office	liquid	−4.7 after consuming OO for 9 wk in Northern Europe subjects vs. baseline; −4.4 after consuming OO for 9 wk in Central Europe subjects vs. baseline	−2.2 after consuming OO for 9 wk in Northern Europe subjects vs. baseline; −3.1 after consuming OO for 9 wk in Central Europe subjects vs. baseline	Only SBP significantly decreased after 9 wk of OO
Rozati, 2015 [54]	41; male and female; 72.0 ± 1; 80 ± 2	overweight and obese	RCT, parallel, USA	12-week intervention with American diet + EVOO or American diet + Control oil (corn oil and soybean oil)	Office	liquid	−6 in EVOO group vs. baseline; no change in control oil group vs. baseline	−3 in EVOO group vs. baseline; −3 in control oil group vs. baseline	Only SBP significantly decreased after 12 wk of EVOO
Venturini, 2015 [55]	102; male and female; 51.4 ± 8.27; NR	metabolic syndrome	RCT, parallel, Brazil	(1) 12-week intervention with 3 g/d of fish oil; (2) 10 mL/d of EVOO at lunch and dinner; (3) fish oil and plus EVOO. (4) control group (usual diet);	Office	liquid	−5 in EVOO group vs. baseline; no change in control oil group vs. baseline	−5 in EVOO group vs. baseline; no change in control oil group vs. baseline	In the OO group both SBP and DBP decreased
Toledo, 2013 [42]	7158, male and female, 66.1 ± 6.1, 30 ± 4 kg/m^2^ (BMI)	at CHD risk and hypertensive	RCT, parallel, multicenter, Spain	4.8-year intervention with (1) MD supplemented with EEVOO, (2) MD supplemented with mixed nuts or (3) control diet (low-fat diet).	Office	liquid	no change in EVOO vs. control	−1.53 in EVOO vs. control; −0.65 in nut group vs. control	Only DBP significantly decreased after 4.5 yr of EVOO
Doménech, 2014 [43]	235, male and female, 66.1 ± 6.1, 78 ± 11	at CHD risk and hypertensive	RCT, parallel, multicenter, Spain	4.8-year intervention with (1) MD supplemented with EVOO, (2) MD supplemented with mixed nuts or (3) control diet (low-fat diet).	24-AMBP	liquid	−2.3 in EVOO vs. baseline; −2.6 in nut groups vs. baseline; +1.7 in the control group vs. baseline	−1.2 in EVOO vs. baseline; −1.2 in nut groups vs. baseline; +0.7 in the control group vs. baseline	SBP and DBP decreased with the MD enriched in EVOO or nut
Martin-Pelàez, 2015 [63]	22, male, 36.0 ± 11.1, 78.5 ± 11.9	healthy	RCT, cross-over, Spain	3-week intervention with 2-week wash-out period, EVOO vs. ROO	Office	liquid	−4.22 in EVOO group vs. baseline	−2.11 in EVOO group vs. baseline	SBP and DBP were significantly reduced only in EVOO
Ceriello, 2104 [41]	22, male and female, NR, 29.1 ± 1.2	diabetics	RCT, parallel, Spain	12-week intervention with MD + EVOO or a control low-fat diet	Office	liquid	no change in EVOO vs. baseline	no change in EVOO vs. baseline	No effect
Passfall, 1993 [56]	10, male and female, age range 40–61, NR	hypertensive	RCT, cross-over, Germany	6-week intervention with 4-week wash-out period, supplementation with OO (9 g) vs. fish oil (9 g)	Office	capsules	no change after OO vs. baseline	no change after OO vs. baseline	No effect

Abbreviations. BMI, body mass index; RCT, randomized controlled trial; EVOO, extra virgin olive oil; OO, olive oil; ROO, refined olive oil; SO, sunflower oil; CLA, conjugated linoleic acid; HOSO, high oleic sunflower oil; SFA, saturated fatty acid; MUFA, monounsaturated fatty acids; PUFA, polyunsaturated fatty acid; CHO, carbohydrate; NR, not reported, AMBP, ambulatory monitoring of blood pressure; HC, hypercolesterolemic; NC, normocholesterolemic; MD, Mediterranean diet, N., number of participants.

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
