# Peer review of "Effects of Olive Oil on Blood Pressure: Epidemiological, Clinical, and Mechanistic Evidence"

_nutrients, 2020, doi:10.3390/nu12061548_

Round 1

Reviewer 1 Report

The current manuscript was well organized and reviewed extensively regarding the effect of olive oil on blood pressure.

After a few minor points might be addressed, the manuscript might be accepted.

  1. In Table 2, the study desing (cross over or parellel, double blind or single blind) should be inserted.
  2. In addition, study site (country) should be added.
  3. The mechanisms should be disccused based on the individula composition. Which compounds are the most contributor to reduce blood pressure? MUFA of oleoreption?

Author Response

Reviewer: The current manuscript was well organized and reviewed extensively regarding the effect of olive oil on blood pressure.

After a few minor points might be addressed, the manuscript might be accepted.

  1. In Table 2, the study desing (cross over or parellel, double blind or single blind) should be inserted.
  2. In addition, study site (country) should be added.

Reply: We thank the Reviewer for the constructive revision of the manuscript and for the careful suggestions that surely will improve the quality of the manuscript.

Regarding the point n.1: in the table 2 for each study the specific experimental design was already reported in the third column. The study site was also already reported but in the text (lanes 259-262).

However, as suggested, we inserted the study site also in the table 2 in the column 3, just after the experimental design.  

Reviewer: The mechanisms should be disccused based on the individula composition. Which compounds are the most contributor to reduce blood pressure? MUFA of oleoreption?

Reply: We again thank the Reviewer for the careful revision of the manuscript. The potential anti-hypertensive mechanisms of EVOO antioxidants are discussed from the lane 441 to 449 and from lane 478 to lane 485, while the mechanisms more specifically related to fat component (oleic acid) are discussed from the lane 463 to 470.

On the bases of reviewed results, it is very difficult to establish which is the component that more efficaciously contribute to blood pressure regulation. Rather EVOO components appear to express their maximal efficacious within the complex of their original matrix. This suggest that the formulation of possible functional products should approximate as much as possible the natural environment in which active molecules are found. Such consideration has been introduced from lane 559 to 564.

Reviewer 2 Report

I commend the authors on this rather exhaustive paper on OO and its cardio-protective effects. A quick search reveals quite a few recent articles on the same subject have been published in recent years. However, this manuscript delves more deeply into the historical tracing of OO production and use from the beginning of civilization. The authors seem to have painstakingly compiled a lot of information into this very detailed paper. Innovative concepts such as the Bubble Map figures add to the appeal of this paper. While I note the mention of low potassium, high sodium, high body weight and lack of activity to be determining factors in the incidence of HTN, there are several others that equally deserve mention such as the effect of magnesium, calcium, folic acid, and vitamins C and D. One suggestion I have is to break down the big paragraphs into a few smaller ones to reduce reader fatigue. Overall, I feel this paper will certainly add to the existing body of literature on OO and its cardiovascular benefits. 

Author Response

I commend the authors on this rather exhaustive paper on OO and its cardio-protective effects. A quick search reveals quite a few recent articles on the same subject have been published in recent years. However, this manuscript delves more deeply into the historical tracing of OO production and use from the beginning of civilization. The authors seem to have painstakingly compiled a lot of information into this very detailed paper. Innovative concepts such as the Bubble Map figures add to the appeal of this paper. While I note the mention of low potassium, high sodium, high body weight and lack of activity to be determining factors in the incidence of HTN, there are several others that equally deserve mention such as the effect of magnesium, calcium, folic acid, and vitamins C and D. One suggestion I have is to break down the big paragraphs into a few smaller ones to reduce reader fatigue. Overall, I feel this paper will certainly add to the existing body of literature on OO and its cardiovascular benefits.

Reply: We thank the Reviewer for the appreciation of the manuscript. This fills us with pride. We followed the very appropriate advice and break down many paragraphs into smaller ones:

We will find now the following paragraphs:

  1. Introduction
  2. Dietary strategies in the management of BP
  3. Olive oil: a succinct history, composition, production and consumption
  4. Olive oil composition
  5. Olive oil and hypertension: evidence from epidemiological studies
  6. Olive oil and hypertension: evidence from clinical trial. Search strategy
  7. Olive oil and hypertension: clinical trial results
  8. Pathogenesis of hypertension
  9. Olive oil anti-hypertensive effects: mechanistic evidence from animal and human studies
  10. Discussion
  11. Conclusions